# Effect of optional biometric parameters in the Kane formula on intraocular lens power calculation

Xiao-Yu Li[1,2], Xuan Liao[1]*, Jia Lin[1], Chang-Jun Lan[3], Qing-Qing Tan[1]

1 Department of Ophthalmology of Affiliated Hospital, North Sichuan Medical College, Nanchong, China,
2 Department of Ophthalmology of Nanchong Central Hospital, The Second Clinical College, North Sichuan Medical College, Nanchong, China, 3 Sichuan Eye Hospital, Chengdu, China

* aleexand@163.com

**Data Availability Statement:** All relevant data are within the paper and its Supporting Information files.

## Abstract

### Purpose

To investigate the effect of the optional biometric parameters lens thickness (LT) and center corneal thickness (CCT) in the Kane formula on intraocular lens (IOL) power calculation.

### Methods

A cross-sectional study included consecutive cataract patients who received uncomplicated cataract surgery with IOL implantation from May to September 2022 were enrolled. The ocular biometric parameters were obtained using IOLMaster 700 and then inputted into online Kane formula calculator. The IOL power was calculated for targeting emmetropia and compared between groups: not omitting (NO) group, omitting LT and CCT (OLC) group, omitting LT (OL) group and omitting CCT (OC) group. Further, according to the axial length (AL), anterior chamber depth (ACD), and mean keratometry ($K_m$), the eyes were divided into three subgroups, respectively.

### Results

1005 eyes of 1005 consecutive patients were included. There was no significant difference in IOL power between NO group and OC group ($P = 0.064$), and the median absolute difference (MedAD) was 0.05D. The IOL power in NO group showed significant differences from OLC group and OL group respectively ($P < 0.001$), and both MedAD values were 0.18D. Among AL subgroups, MedAD ranged from 0.06D to 0.35D in short eyes. Among ACD subgroups, the above values ranged from 0.06D to 0.23D in shallow ACD subgroup. Among $K_m$ subgroups, these values ranged from 0.05D to 0.31D in steep $K_m$ subgroup.

### Conclusion

The optional biometric parameter CCT has no effect on the calculation results of the Kane formula, whereas the parameter LT has a great influence on the Kane formula results for the IOL power calculation in cataract patients with short AL, shallow ACD and steep $K_m$.

**Funding:** Supported by the Project of the Strategic Cooperation of City and College (No. 22SXFWDF0003).The funders had no role in study design, data collection and analysis, decision to publish, or preparation of the manuscript.

**Competing interests:** The authors have declared that no competing interests exist.

## Introduction

The prediction accuracy of intraocular lens (IOL) refractive power has become the key to achieving good postoperative vision [1]. This also prompted the IOL power calculation formulas to update constantly, from the historical or refraction-based formulas, empirical formulas and vergence formulas, to the current artificial intelligence based formulas and ray tracing calculation formulas [2].

A large number of clinical studies have found that the Kane formula, a new generation of IOL calculation formula, was proved to be the most accurate formula in cataract patients with different axial length (AL) spectra [3–5]. Kane formula calculates IOL refractive power using biological sex (BS) and ocular biometric parameters, included AL, anterior chamber depth (ACD), keratometry readings (K), lens thickness (LT) and center corneal thickness (CCT). Although adding LT and CCT improves the prediction, they are set as optional variables. This allows the formula to be used in cases where LT and CCT measurements are not performed due to equipment limitations, thus broadening the application range of the formula.

There is still a lack of studies on the role of the optional biometric parameters in the Kane formula for IOL power calculation. This study aims to compare the IOL power calculation by selecting all biometric parameters or omitting optional biometric parameters in the Kane formula, so as to explore the influence of optional biometric parameters of the Kane formula on the IOL power calculation.

## Materials and methods

### Patients and groupings

A cross-sectional study included consecutive cataract patients who received uncomplicated cataract surgery with IOL implantation in the Affiliated Hospital of North Sichuan Medical College from May to September 2022. This study followed the Declaration of Helsinki and was approved by the Ethics Committee of the Affiliated Hospital of North Sichuan Medical College (2021ER151-1). The informed consents were signed. This study has been registered in the Chinese Clinical Trial Registry (ChiCTR2200059168).

Inclusion criteria were as follows: (1) patients with age-related cataract or high myopes with cataract, and Emery nuclear grade II ~ IV; (2) preoperative corneal astigmatism less than 2.0D; (3) uncomplicated cataract surgery and IOL implantation were performed; (4) complete preoperative data; (5) implantation of ZCB00 IOL (Advanced Medical Optics, USA). The exclusion criteria were: (1) patients who could not be measured by IOLMaster 700 (Carl Zeiss Meditec, Jena, Germany); (2) patients with any corneal diseases, such as keratoconus, pterygium, or corneal scarring of any etiology; (3) patients with fundus lesions; (4) patients with a history of ocular surgery or ocular trauma. If patients had underwent bilateral surgeries, only one eye included was chosen randomly.

Groups according to the biometric parameters were involved in the Kane formula: (1) not omitting (NO) group; (2) omitting LT and CCT (OLC) group; (3) omitting LT (OL) group; (4) omitting CCT (OC) group. Patients were subgrouped according to the preoperative biometric parameters measured by IOLMatser 700. According to Kane [6] AL classification method, eyes were divided into: short AL subgroup (AL $\leq$ 22.00mm), normal AL subgroup (22.00mm<AL<26.00mm), and long AL subgroup (AL $\geq$ 26.00mm). According to Hipolito [7] ACD classification method, eyes were divided into: shallow ACD subgroup (ACD $\leq$ 3.00mm), normal ACD subgroup (3.00mm < ACD < 3.50mm), and deep ACD subgroup (ACD $\geq$ 3.50mm). According to Kim [8] $K_m$ classification method, eyes were divided into: flat $K_m$ subgroup ($K_m \leq$ 42.00D), normal $K_m$ subgroup (42.00D < $K_m$ < 46.00D), and steep $K_m$ subgroup ($K_m \geq$ 46.00mm).

## Examinations and surgical procedures

All patients underwent routine ophthalmic examination before surgery, including slit-lamp, best corrected visual acuity, intraocular pressure, corneal endothelium, B scan and fundus examination. A professional was assigned to measure ocular biometric parameters including AL, keratometry ($K_1$ and $K_2$), mean keratometry ($K_m$), ACD, LT and CCT using a swept-source optical coherence tomography (SS-OCT) biometer IOLMaster 700 (software version 1.88). All scans were carried out through the macular fovea under fixation station, and measurements with exclamation marks or asterisks were eliminated. The patients were operated by a single senior surgeon. The uncomplicated phacoemulsification and IOL implantation were performed under topical anesthesia. Through a 2.4 mm temporal clear corneal incision, a 5.5 ~ 6.0 mm continuous curvilinear capsulorhexis was made followed by phacoemulsification, and an IOL was then implanted into the capsular bag.

The ocular biometric parameters were inputted into the online Kane formula calculator (https://www.iolformula.com/). Using the optimized A-constant for the Kane formula of 119.36 and targeting emmetropia, the IOL power were calculated and compared between different combinations of mandatory or alternative biometric parameters (LT and CCT). Calculation difference (CD) is the calculation result of IOL power in NO group minus the result in OLC group (or OC group or OL group). The percentage of eyes within ±0.25D of CD (POCD ±0.25D) were calculated for each group. Absolute calculation difference (AD) is the absolute value of the difference of calculation result of IOL power in NO group minus the result in OLC group (or OC group or OL group), and the median absolute calculation difference (MedAD) is the median of them. The smaller the CD (or AD or MedAD) is, the larger value of POCD±0.25D will be, indicating that the optional biometric parameters have less influence on the accuracy of IOL power calculation.

## Statistical analysis

SPSS 26.0 software was used for statistical analysis. *Kolmogorov-Smirnov* test was used to test the normality of the data. The data conforming to the normal distribution were represented by mean ± standard deviation (SD), while the skewed data were represented by median and quartile [$M (Q_1, Q_3)$]. Wilcoxon rank-sum test was used to compare the IOL power calculation results among groups. Kruskal-Wallis H test was performed to compare the values of AD among subgroups. Pearson's Chi-square test or Fisher's exact test was used to compare the values of POCD±0.25D among subgroups. The difference was considered statistically significant if *P* value was less than 0.05. The sample size required for this study was calculated using PASS 15.0 software, with significance level of 0.05 and a power of 90%. According to the software calculation results, the total sample size of 532 eyes would be required. The total sample size of this study was 1005 cases, so it had a certain reference value for statistical treatment.

## Results

A total of 1005 eyes (504 right and 501 left eyes) of 1005 consecutive patients (602 females and 403 males) were recruited and biometric parameters were presented in Table 1. The mean age of the patients was 65.36±11.10 years (ranged 30–91 years), the mean AL was 24.55±3.17 mm (ranged 20.09–35.00 mm), the mean $K_m$ was 44.14±1.51D (ranged 37.72–49.16 D), and the mean ACD was 3.03±0.50 mm (ranged 1.52–4.47 mm). The eyes were divided as follow: AL subgroups, including short ($n = 228$), normal ($n = 527$) and long ($n = 250$); ACD subgroups, including shallow ($n = 473$), normal ($n = 339$) and deep ($n = 193$); and $K_m$ subgroups, including flat ($n = 72$), normal ($n = 821$) and steep ($n = 112$).

**Table 1. Ocular biometry data.**

| | Mean ± SD | Range |
|---|---|---|
| Age (years) | 65.36±11.10 | 30.00–91.00 |
| AL (mm) | 24.55±3.17 | 20.09–35.00 |
| $K_1$ (D) | 43.73±1.51 | 37.22–48.68 |
| $K_2$ (D) | 44.54±1.54 | 38.22–49.67 |
| $K_m$ (D) | 44.14±1.51 | 37.72–49.16 |
| ACD (mm) | 3.03±0.50 | 1.52–4.47 |
| LT (mm) | 4.41±0.46 | 2.50–5.83 |
| CCT (mm) | 539.16±33.65 | 420.00–650.00 |

AL, axial length; $K_1$, flat keratometry; $K_2$, steep keratometry; $K_m$, mean keratometry; ACD, anterior chamber depth; LT, lens thickness; CCT, center corneal thickness.

The results of IOL power calculation in the NO group were compared with those in OLC group, OC group and OL group, as shown in Table 2 and Fig 1. There were no significant differences in IOL power between NO group and OC group ($P = 0.064$), and the value of MedAD was 0.05D. However, the IOL power of NO group was significantly different from OLC group and OL group respectively ($P<0.001$), and both MedAD values were 0.18D.

The AD values based on AL subgroups were shown in Fig 2A and Table 3. The results showed that AD values of OLC group and OL group in patients with short eyes were highest, in which the MedAD values were both 0.35D. However, the AD values of the long eyes were lowest, and the MedAD values were 0.08D. There were no significant differences in the AD values of OC group among the AL subgroups ($P = 0.119$). The AD values of ACD subgroups were shown in Fig 2B and Table 4. The optional biometric parameters had the greatest influence on IOL power calculation in shallow ACD (MedAD ranged from 0.06D to 0.23D), while the optional biometric parameters had the least influence on deep ACD (MedAD ranged from 0.04D to 0.10D). The AD values of $K_m$ subgroups were shown in Fig 2C and Table 5. The results indicated that AD values of OLC group and OL group in the steep $K_m$ subgroup were highest, in which the MedAD values were both 0.31D, and the lowest MedAD values were 0.05D in flat $K_m$ subgroup. The differences of AD values in OC group among $K_m$ subgroups did not reach statistical significance ($P = 0.518$).

In addition, the POCD±0.25D values among different subgroups were shown in Fig 3 and Tables 3–5. For AL, the POCD±0.25D values of the short-eye subgroup were lowest, in which the value was 97.37% in OC group, and 34.65% in both OLC and OL groups. However, the POCD±0.25D values of the long eyes were highest, which was 100% in OC group, and 86.80% in both OLC group and OL group. For ACD, the POCD±0.25D values of the shallow ACD subgroup were lowest, which were 53.28% in both OLC and OL group; The POCD±0.25D values of the deep ACD were highest, which were 81.87% in both OLC and OL group. For $K_m$,

**Table 2. AD values of OLC, OC and OL group compared to NO group.**

| Group | Parameters Used | AD [$M (Q_1, Q_3)$] (D) | Z | P |
|---|---|---|---|---|
| OLC | BS, AL, K, ACD | 0.18(0.08, 0.32) | -15.443 | < 0.001 |
| OC | BS, AL, K, ACD, LT | 0.05(0.02, 0.08) | -1.852 | 0.064 |
| OL | BS, AL, K, ACD, CCT | 0.18(0.08, 0.32) | -15.443 | < 0.001 |

BS, biological sex; AL, axial length; K, keratometry; ACD, anterior chamber depth; LT, lens thickness; CCT, center corneal thickness.

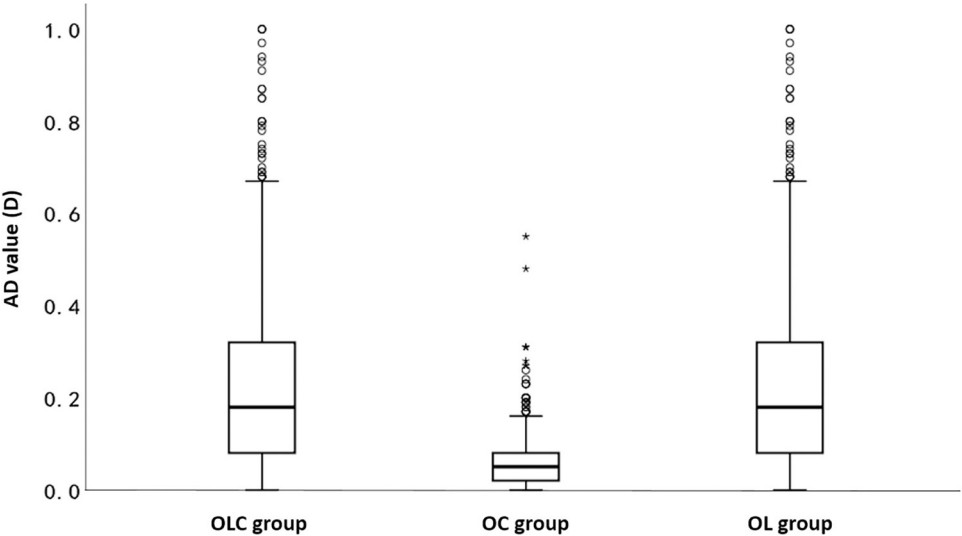

**Fig 1. Box-plot of AD values of OLC group, OC group and OL group.**

the POCD±0.25D values of the steep ACD subgroup were lowest, which were 43.75% in both OLC and OL group; The POCD±0.25D values of the flat $K_m$ were highest, which were 80.56% in OLC group and OL group. Moreover, no significant differences were found in the POCD ±0.25D values of OC group among the subgroups which were divided by ACD and $K_m$ ($P$ = 0.067, 0.640, respectively).

## Discussion

With the advances of IOL calculation formulas, the prediction accuracy hasgreatly improved. According to the number of preoperative biometric parameters, early IOL calculation formulas can be divided into 2-variable formulas (Hoffer Q [9], SRK/T [10] and Holladay 1 [11]) and 3-variable formulas (Haigis [12]). The 2-variable formula predicted the effective lens position (ELP) according to AL and K, while the 3-variable formula considered the effects of AL and preoperative ACD on ELP. The advent of optical coherence biometers provided the opportunity to measure ocular biometric parameters more precisely. The 5-variable formula (Barrett Universal II), 6-variable formula (Kane), and 7-variable formula (Holladay 2) have also appeared successively [13]. However, LT, CCT and white to white (WTW) are usually used as optional biometric parameters in the calculation formulas. Theoretically, more biometric parameters involved in the calculation of IOL power can provide more accurate results, and its calculation result will vary with the different biometric parameters involved. To our knowledge, this study is the first to evaluate the effect of optional biometric parameters in the Kane formula on IOL power calculation in different selections of AL, ACD and $K_m$.

Kane formula is based on theoretical optics combines regression research with artificial intelligence technology, and uses efficient cloud-based algorithm to obtain calculation results with high accuracy [14]. The mandatory biometric parameters BS, AL, ACD, K and optional biometric parameters LT and CCT can be used to calculate the IOL refractive power. Although authors of the Kane formula suggested the convenience of using optional biometric parameters LT and CCT, the strategy of mandatory biometric parameters was widely used due to its high accuracy. Connell *et al.* [15] included 846 cataract patients implanted with SN60WF, and compared the accuracy of Kane, Hill-RBF, Olsen, BU II, Haigis, SRK/T, Hoffer Q, Holladay 1 and

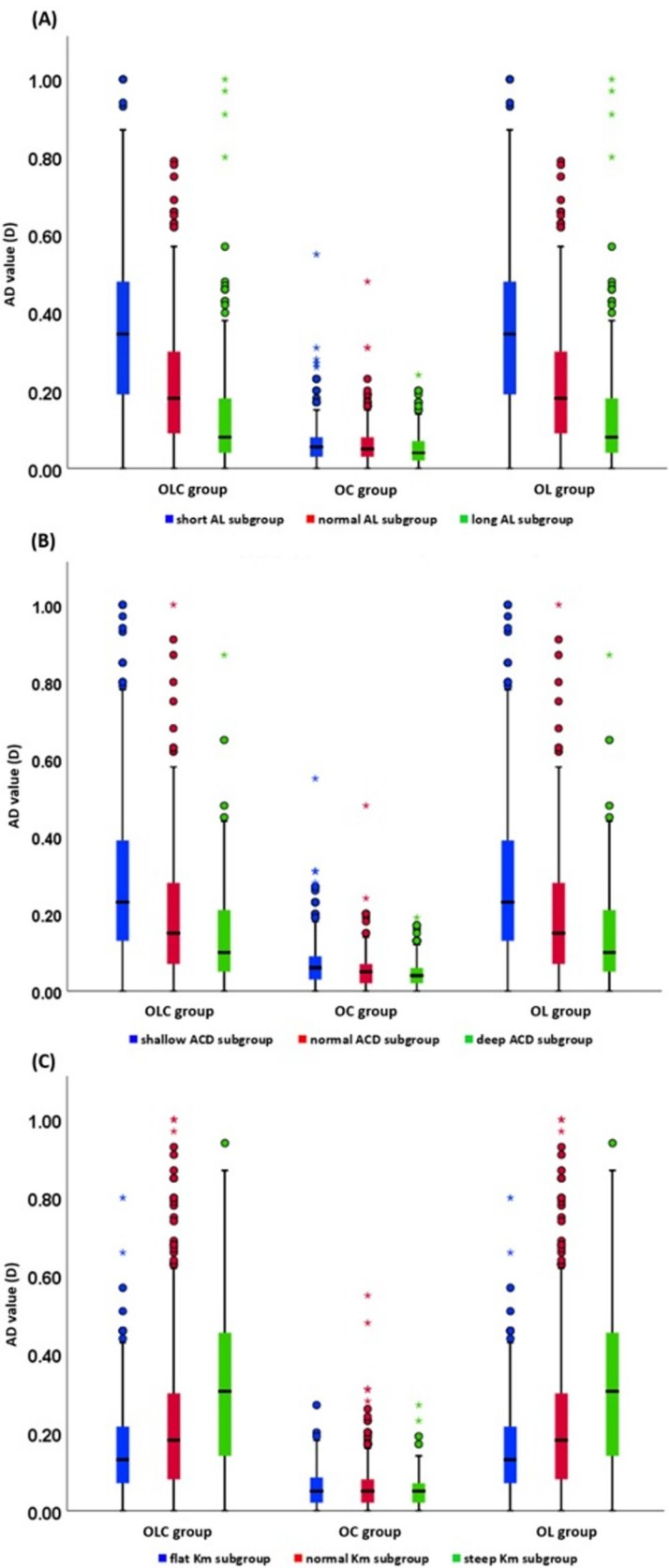

**Fig 2. Box-plot of AD values in different optional parameter groups.** (A, AL subgroups; B, ACD subgroups; C, $K_m$ subgroups).

Holladay 2 formulas. The results showed that Kane formula had the highest accuracy among all patients when mandatory biometric parameters (BS, AL, ACD and K) were entered. However, when biometric instruments or ocular conditions were limited in clinical practice, only partial optional biometric parameters can be measured, which affects the calculation of IOL power.

Therefore, by comparing the IOL power results of the Kane formula when all biometric parameters were included with the results when biometric parameters were changed, this study found that there was no significant difference between NO group and OC group ($P = 0.064$), and MedAD value was 0.05D. The IOL power in NO group had significant differences with OLC group and OL group respectively ($P < 0.001$), and the both MedAD values were 0.18D. Given this, LT plays an important role in the IOL power calculation of the Kane formula. It is speculated that the reason may be that the Kane formula takes into account the influence of ACD on postoperative ELP, and it is logical to assume that LT has a certain effect on ACD. Due to LT thickens with age, LT is usually negatively correlated with ACD, and the combination of LT and ACD can improve the prediction of postoperative ELP [16]. It is worth noting that the OLC group and the OL group have the same IOL power calculation results, that is, adding optional biometric parameter CCT alone has no effect on the calculation results of the Kane formula. Thus, it is necessary to calculate the IOL power combined with LT. Moreover, due to the artificial intelligence algorithm of the Kane formula itself, regression analysis of 30,000 eyes in the database may result in CCT as a predictor of insignificant importance.

In addition, this study further subgrouped subjects according to AL, ACD and $K_m$, and explored the influence of optional biometric parameters of the Kane formula on IOL power calculation in each subgroup. The analysis showed that short AL, shallow ACD and steep $K_m$ had significant effects on the results of the Kane formula. Given that in these cases patients need high power IOL implantation, any little parameter change may have a great impact on the outcomes, and therefore the optional biometric parameters are particularly important for the calculation in the Kane formula. Moreover, these abnormal biometric parameters often have a great impact on the IOL power calculation and the prediction error occurs, so all biometric parameters should be considered to help improve the prediction accuracy of the Kane

**Table 3. AD values and POCD±0.25D values in AL subgroups.**

| | Short AL (n = 228) | Normal AL (n = 527) | Long AL (n = 250) | | |
|---|---|---|---|---|---|
| AD [$M$ ($Q_1$, $Q_3$)] (D) | | | | $H$ | $P$ |
| OLC | 0.35(0.19, 0.48) | 0.18(0.09, 0.30) | 0.08(0.04, 0.18) | 197.883 [a] | < 0.001 |
| OC | 0.06(0.03, 0.08) | 0.05(0.03, 0.08) | 0.04(0.02, 0.07) | 4.262 [a] | 0.119 |
| OL | 0.35(0.19, 0.48) | 0.18(0.09, 0.30) | 0.08(0.04, 0.18) | 197.883 [a] | < 0.001 |
| POCD±0.25D | | | | $\chi^2$ | $P$ |
| OLC | 34.65% | 67.17% | 86.80% | 144.980 [b] | < 0.001 |
| OC | 97.37% | 99.43% | 100.00% | 8.353 [c] | 0.007 |
| OL | 34.65% | 67.17% | 86.80% | 144.980 [b] | < 0.001 |

[a] Kruskal-Wallis H test

[b] Pearson's Chi-square test

[c] Fisher's exact test.

**Table 4. AD values and POCD±0.25D values in ACD subgroups.**

| | Shallow ACD (n = 473) | Normal ACD (n = 339) | Deep ACD (n = 193) | | |
|---|---|---|---|---|---|
| AD [$M$ ($Q_1$, $Q_3$)] (D) | | | | H | P |
| OLC | 0.23(0.13, 0.39) | 0.15(0.07, 0.28) | 0.10(0.05, 0.21) | 82.971 [a] | < 0.001 |
| OC | 0.06(0.03, 0.09) | 0.05(0.02, 0.07) | 0.04(0.02, 0.06) | 13.508 [a] | < 0.001 |
| OL | 0.23(0.13, 0.39) | 0.15(0.07, 0.28) | 0.10(0.05, 0.21) | 82.971 [a] | < 0.001 |
| POCD±0.25D | | | | $\chi^2$ | P |
| OLC | 53.28% | 70.80% | 81.87% | 57.422 [b] | < 0.001 |
| OC | 98.31% | 99.71% | 100.00% | 5.323 [c] | 0.067 |
| OL | 53.28% | 70.80% | 81.87% | 57.422 [b] | < 0.001 |

[a] Kruskal-Wallis H test

[b] Pearson's Chi-square test

[c] Fisher's exact test.

formula. Previous studies have also investigated the influence of optional biometric parameters of some formulas on IOL power calculation. Vega [17] continuously enrolled 501 cataract patients to evaluate the effect of optional biometric parameters of BU II formula on IOL power calculation. Their results showed that in short AL subgroup, there were significant clinical differences in the results of IOL power calculation between using partial biometric parameters and using all biometric parameters in BU II formula, while in normal AL subgroup and long AL subgroup, optional biometric parameters had little influence on IOL power calculation. Srivannaboon *et al.* [18] analyzed the influence of LT in Holladay 2 formula on the accuracy of IOL power calculation in 143 cataract patients, and found that there was no statistical difference in IOL power calculation results of Holladay 2 formula with and without LT in all AL subgroups. However, only 15 short eyes were included in the above study, and AL was measured by optical biometrics but LT was measured by ultrasound biometrics. Studies have found that the LT measured by optical biometrics and ultrasonic biometrics was not interchangeable, and the difference in measurement might affect the research results [19]. Thus, the results need to be further determined. In addition, Taroni *et al.* [20] conducted a study of 169 cataract patients and found that the use of the optional biometric parameter LT in Kane,

**Table 5. AD values and POCD±0.25D values in $K_m$ subgroups.**

| | Flat $K_m$ (n = 72) | Normal $K_m$ (n = 821) | Steep $K_m$ (n = 112) | | |
|---|---|---|---|---|---|
| AD [$M$ ($Q_1$, $Q_3$)] (D) | | | | H | P |
| OLC | 0.13(0.07, 0.22) | 0.18(0.08, 0.30) | 0.31(0.14, 0.46) | 28.261 [a] | < 0.001 |
| OC | 0.05(0.02, 0.09) | 0.05(0.02, 0.08) | 0.05(0.02, 0.07) | 1.317 [a] | 0.518 |
| OL | 0.13(0.07, 0.22) | 0.18(0.08, 0.30) | 0.31(0.14, 0.46) | 28.261 [a] | < 0.001 |
| POCD±0.25D | | | | $\chi^2$ | P |
| OLC | 80.56% | 66.14% | 43.75% | 30.183 [b] | < 0.001 |
| OC | 98.61% | 99.15% | 99.11% | 0.898 [c] | 0.640 |
| OL | 80.56% | 66.14% | 43.75% | 30.183 [b] | < 0.001 |

[a] Kruskal-Wallis H test

[b] Pearson's Chi-square test

[c] Fisher's exact test.

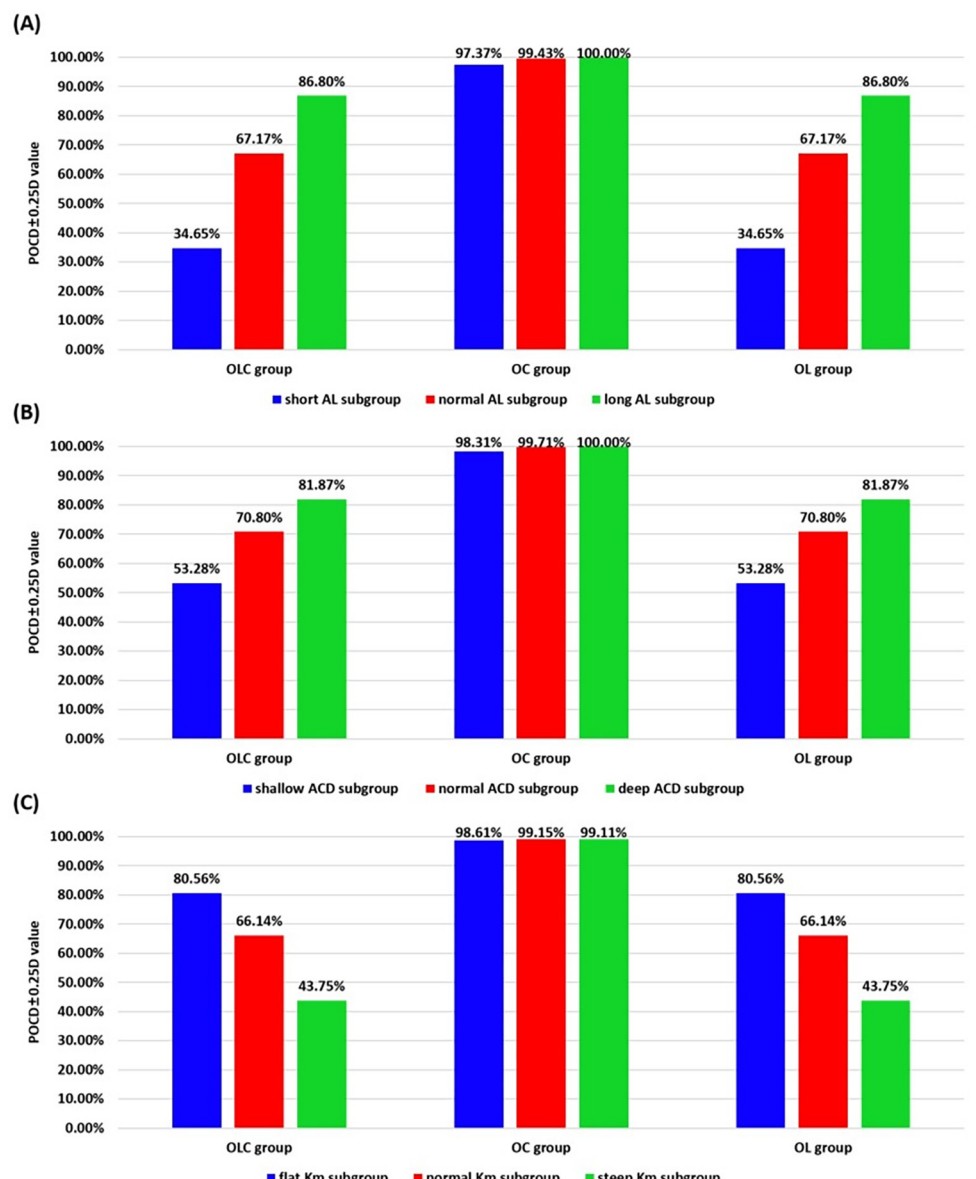

**Fig 3. Histogram of POCD±0.25D values in different optional parameter groups.** (A, AL subgroups; B, ACD subgroups; C, $K_m$ subgroups).

BU II, EVO and RBF formula did not improve the accuracy of the calculation formula. However, a relatively small number of short eyes were included in his study, and no subgroup analysis was conducted for the short AL group.

There are still some limitations in this study: the sample size of the flat $K_m$ subgroup (n = 72) and the steep $K_m$ subgroup (n = 112) may still be insufficient to show statistical effects. Due to methodological limitations of this study, postoperative follow-up and refractive status data were not available. Therefore, further clinical studies are needed to evaluate the contribution of optional biometric parameters to the accuracy of the Kane formula.

In summary, the optional biometric parameter CCT has no effect on the calculation results of the Kane formula, whereas the optional biometric parameter LT has a great influence on

cataract patients with short AL, shallow ACD and steep Km. Under these conditions, it is recommended that LT should be employed when using the Kane formula for IOL power calculation.

## Supporting information

**S1 Data.**
(XLSX)

## Author Contributions

**Conceptualization:** Xiao-Yu Li, Xuan Liao, Jia Lin, Chang-Jun Lan, Qing-Qing Tan.

**Data curation:** Xiao-Yu Li.

**Formal analysis:** Xiao-Yu Li, Xuan Liao, Jia Lin, Chang-Jun Lan, Qing-Qing Tan.

**Funding acquisition:** Xiao-Yu Li.

**Investigation:** Xiao-Yu Li.

**Methodology:** Xiao-Yu Li, Xuan Liao, Jia Lin, Chang-Jun Lan, Qing-Qing Tan.

**Project administration:** Xiao-Yu Li, Xuan Liao, Jia Lin, Chang-Jun Lan, Qing-Qing Tan.

**Resources:** Xiao-Yu Li.

**Software:** Xiao-Yu Li.

**Supervision:** Xuan Liao, Chang-Jun Lan, Qing-Qing Tan.

**Validation:** Xiao-Yu Li.

**Visualization:** Xiao-Yu Li.

**Writing – original draft:** Xiao-Yu Li.

**Writing – review & editing:** Xuan Liao, Jia Lin, Chang-Jun Lan, Qing-Qing Tan.

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
