## [Decision Letter · Decision Letter 0]

28 Apr 2023

PONE-D-23-03187Effect of optional biometric parameters in the Kane formula on intraocular lens power calculationPLOS ONE

Dear Dr. Liao,

Thank you for submitting your manuscript to PLOS ONE. After careful consideration, we feel that it has merit but does not fully meet PLOS ONE’s publication criteria as it currently stands. Therefore, we invite you to submit a revised version of the manuscript that addresses the points raised during the review process.

We look forward to receiving your revised manuscript.

Kind regards,

Manuel Garza León

Academic Editor

PLOS ONE

Journal Requirements:

5. Please upload a new copy of Figures 1,2 and 3 as the detail is not clear. Please follow the link for more information: " ext-link-type="uri" xlink:type="simple">https://blogs.plos.org/plos/2019/06/looking-good-tips-for-creating-your-plos-figures-graphics/"
https://blogs.plos.org/plos/2019/06/looking-good-tips-for-creating-your-plos-figures-graphics/

Reviewers' comments:

Reviewer's Responses to Questions

**Comments to the Author**

1. Is the manuscript technically sound, and do the data support the conclusions?

Reviewer #1: Yes

Reviewer #2: Yes

2. Has the statistical analysis been performed appropriately and rigorously? 

Reviewer #1: Yes

Reviewer #2: Yes

3. Have the authors made all data underlying the findings in their manuscript fully available?

Reviewer #1: Yes

Reviewer #2: Yes

4. Is the manuscript presented in an intelligible fashion and written in standard English?

Reviewer #1: Yes

Reviewer #2: Yes

5. Review Comments to the Author

Reviewer #1: With the advances of IOL calculation formulas, the prediction accuracy has been greatly improved.Patients are also more demanding with refractive results.The cloud-based algorithm of Kane formula is perfect to make statistical more accurate calculations and enhance results .The more reliable data we can introduce , in any formula, not only Kane , the accurate the refractive results we will achieve .This is well known since Haigis Formulas , the first in the world to consider ACD and LENS thickness.CCT stills look at a parameter with no influence at all in any formula. The subgroup analisis that the authors have done in this paper is really clear and well developed.It finally shows what we all know about short eyes with small ACD and stepped K. The analysis showed that short AL, shallowACD and steep Km had significant effects on the results of the Kane formula and will do in any formula.The ELP looks better predictable in Kane formula but stills a great black hole in our practice . The only weakness of the article is the lack of follow up data and refractive status .

Reviewer #2: The study is very interesting and has a good sample size to corroborate its results. The results are valuable and important for clinical practice.

The necessary changes I consider are the following:

Review writing and spelling. Some commas are missing.

It does not state the number of patients in the subgroups. It would be worth creating a table that shows the values of each variable in each group with their distribution.

I suggest making a statistical comparison between the subgroups.

The study shows that the CCT has no effect on the iol calculation, but in the conclusions, it states that both optional parameters have an influence on the Kane formula. This conclusion is opposite to its results.

6. PLOS authors have the option to publish the peer review history of their article (what does this mean?). If published, this will include your full peer review and any attached files.

Reviewer #1: **Yes: **Maria Julia Zunino

Reviewer #2: No

---

## [Author Response · Author response to Decision Letter 0]

12 Jun 2023

Dear editor and reviewers,

Thanks for your efforts in reviewing our manuscript titled “Effect of optional biometric parameters in the Kane formula on intraocular lens power calculation” (ID: PONE-D-23-03187R1), and for providing many helpful comments and suggestions, which will all prove invaluable in the revision and improvement of our manuscript, as well as in guiding our research in the future. 

We have studied your letter and the reviewers' comments point by point and revised the manuscript accordingly. The responses to the reviewer's comments are presented following. All authors have approved the response letter and the revised version of the manuscript.

Thank you again for your valuable comments and suggestions. We look forward to hearing from you soon in due course. 

Yours sincerely,

Xuan Liao

Email: aleexand@163.com

Reviewer #1: With the advances of IOL calculation formulas, the prediction accuracy has been greatly improved. Patients are also more demanding with refractive results. The cloud-based algorithm of Kane formula is perfect to make statistical more accurate calculations and enhance results. The more reliable data we can introduce, in any formula, not only Kane , the accurate the refractive results we will achieve .This is well known since Haigis Formulas , the first in the world to consider ACD and LENS thickness. CCT stills look at a parameter with no influence at all in any formula. The subgroup analisis that the authors have done in this paper is really clear and well developed. It finally shows what we all know about short eyes with small ACD and stepped K. The analysis showed that short AL, shallow ACD and steep Km had significant effects on the results of the Kane formula and will do in any formula. The ELP looks better predictable in Kane formula but stills a great black hole in our practice. The only weakness of the article is the lack of follow up data and refractive status.

Response: Many thanks for your recognition and valuable advice. However, the present study aimed to compare the IOL power calculation results of Kane formula with all biometric parameters selected or optional biometric parameters omitted, to explore the influence of optional biometric parameters LT and CCT in the Kane formula on IOL power calculation. Due to methodological limitations of this study, postoperative follow-up and refractive status data were not available, so we unable to assess how the changes in IOL power calculation influenced the accuracy of the Kane formula. 

Although investigating the accuracy of Kane's formula was beyond the scope of our study, there is no denying the importance of the reviewer's opinion that further clinical studies are needed to assess the contribution of optional biometric parameters to the accuracy of Kane's formula. This has been mentioned in the manuscript as a limitation of the present study.

Reviewer #2: The study is very interesting and has a good sample size to corroborate its results. The results are valuable and important for clinical practice. The necessary changes I consider are the following: 1. Review writing and spelling. Some commas are missing. 2. It does not state the number of patients in the subgroups. It would be worth creating a table that shows the values of each variable in each group with their distribution.

3. I suggest making a statistical comparison between the subgroups.

4. The study shows that the CCT has no effect on the iol calculation, but in the conclusions, it states that both optional parameters have an influence on the Kane formula. This conclusion is opposite to its results.

Responses:

1. Thanks for your suggestion and we have carefully reviewed the writing and spelling and added the missing commas in the revised manuscript.

2. We appreciate you pointing out this issue in our manuscript and have modified the manuscript accordingly. We have created tables to show the number of patients in the subgroups and the values of each variable in each group with their distribution. Please see tables 3-5.

3. We have made the changes to the manuscript to address this comment. We have conducted statistical comparison on these subgroups and created tables to display the results. Please see tables 3-5 and the results section of the manuscript. Many thanks.

4. Thank you for your careful checking. We have modified the conclusions to make them consistent with the results.

---

## [Editor Report · Decision Letter 1]

10 Jul 2023

Effect of optional biometric parameters in the Kane formula on intraocular lens power calculation

PONE-D-23-03187R1

Dear Dr. Liao,

We’re pleased to inform you that your manuscript has been judged scientifically suitable for publication and will be formally accepted for publication once it meets all outstanding technical requirements.

Kind regards,

Manuel Garza León

Academic Editor

PLOS ONE
---

## [Editor Report · Acceptance letter]

15 Aug 2023

PONE-D-23-03187R1 

Effect of optional biometric parameters in the Kane formula on intraocular lens power calculation 

Dear Dr. Liao:

I'm pleased to inform you that your manuscript has been deemed suitable for publication in PLOS ONE. Congratulations! Your manuscript is now with our production department. 

Kind regards, 

on behalf of

Dr. Manuel Garza León 

Academic Editor

PLOS ONE